# The Potential of Human–Horse Attachment in Creating Favorable Settings for Professional Care: A Study of Adolescents’ Visit to a Farm

**DOI:** 10.3390/ani10091707

**Published:** 2020-09-21

**Authors:** Erna Törmälehto, Riikka Korkiamäki

**Affiliations:** Faculty of Social Sciences, Pori Unit, Tampere University, PL/Po-Box 185, FI-28101 Pori, Finland; riikka.korkiamaki@tuni.fi

**Keywords:** adolescent, attachment, equine-assisted therapies, human-horse relations, professional care

## Abstract

**Simple Summary:**

Although previous research has shown that the features of an attachment bond are fulfilled in human–dog dyads, it is still unclear if people may have horses as attachment figures. This paper examines whether adolescents’ contact with horses can result in attachment bonds and whether these forms of attachment hold any potential for motivating adolescents in professional care. The paper is based on a time-limited session where six girls and three boys, aged 16–17, were observing and interacting with free-roaming horses at a farm. The results suggest that the presence of horses may activate the development of attachment in adolescents in the same way people do. The four criteria of an attachment bond—proximity maintenance, safe haven, secure base, and separation distress—may be fulfilled in the interaction between a human and a horse. This promotes a therapeutically favorable environment for professional care that is appealing, unique, and accurate. The study provides insights into horses’ therapeutic potential in dealing with adolescents in need of professional therapy or care.

**Abstract:**

Previous research has shown features of an attachment bond to be fulfilled in, for instance, human–dog dyads; however, there is a considerable lack of research on the potential attachment in human–horse relationships. Employing Bowlby’s criteria of an attachment bond and Pierce’s model of therapeutically powerful activity, this article studies whether short-term exposure to horses brings about elements of emerging attachment for adolescents and if this interaction holds potential in creating a favorable early-stage setting for professional care. It draws from group discussions carried out with nine 16–17-year-old adolescents who participated in an EASEL (Equine-Assisted Social and Emotional Learning) session when visiting a farm with a youth worker. A qualitative content analysis of the discussions revealed that some characteristics of the four principal criteria of an attachment bond—proximity maintenance, safe haven, secure base, and separation distress—were identifiable in the adolescents’ expressed experiences of observing and interacting with horses. Moreover, the three main sources of therapeutic power—appeal, accuracy, and intactness—intersected with the emerging development of the adolescents’ attachment to horses. Additionally, space for self-reflection was enhanced by the presence of the horses. The study offers insights into the potential of human–horse attachment in dealing with adolescents with and without special needs for various therapy and care purposes.

## 1. Introduction

Research has pointed out the significance of animals to humans in providing their owners with companionship, friendship, a sense of empathy, the development of emotional bonds, social support, and feelings of safety [1,2,3,4,5]. Particularly for children and young people, animals act as playmates and confidants [6], sometimes developing an even higher level of significance in their lives than their human companions [7]. Cassells [7] suggests that young adolescents who engage with animals at home derive more “satisfaction” and engage in fewer conflicts with their pets than with their siblings. The inclusion of animals in young people’s lives is said to provide a safe haven and a secure base, and to offer a source of comfort during the challenging and complex processes of adolescence [8,9,10]. Animals represent reality—they eat, play, and show emotions—while maintaining a safe psychological distance, allowing a young person to work through their issues with affectionate compassion that is not intrusive [11]. Animals offer friendship via non-verbal cues, which for many adolescents is an easier form of communication than the verbal communication typical for human interaction [12]. 

For the wellbeing and positive development of children and adolescents, building emotional bonds that support a sense of closeness, security, and confidence is essential [13]. This is often explained through Bowlby’s [14] attachment theory, describing how humans strive to have close and meaningful relationships with others [13]. According to Bowlby’s theory, attachment is a deep and enduring emotional bond that connects one person to another across time and space [14,15,16]. It is characterized by specific behaviors in children and adolescents, such as seeking proximity to an attachment figure and feeling distressed in the absence of one, and by certain conditions provided by the environment, such as safety and security [16,17]. Importantly, an attachment figure acts for a child as a secure base for exploring the world, thus establishing an attachment relationship that may function as a prototype for future social relationships [13]. 

While attachment is usually described as an emotional bond between humans, both in mother–infant relationships [14] and in relationships among adults [18,19,20], it has been suggested that non-humans also form attachments [21]. Animals, especially mammals, have been shown to have emotions and close relationships with each other, as well as needing to access social contacts [21]. It has also been claimed that through their sociability and emotional capabilities, animals form social bonds not only with other animals but also with humans [18,22,23,24]. Hence, animals may offer children and young people a way to develop a relationship that affords them the attachment they crave. Several elements of Bowlby’s attachment theory, such as “safe havens”, “affect mirroring”, “reflective functioning” and “non-verbal communication”, have been applied to various animal-assisted interventions [8]. Hence, there is reason to believe that horses, as well as other animals, can provide the therapeutic space for adolescents to develop attachment-driven confidence and security that help them develop relationships with professional care workers [25]. 

This article studies the potential development of an attachment bond in human–horse dyads. Previous research has shown certain features of a typical attachment bond to be fulfilled in human–dog companionship [26], but there is a considerable lack of research on the attachment in human–horse relationships. Following the call by Bachi [8] and Vincent and Farkas [27], who argue that human–horse dyads should also be reviewed in the context of attachment theory, the aim of this qualitative case study is to examine if even a short-term exposure to horses may bring about elements of emerging attachment for adolescents. Specifically, we will look at the four elements of attachment bond named as principal features by Bowlby: (1) proximity maintenance, (2) safe haven, (3) secure base, and (4) separation distress [15]. Furthermore, it is considered whether horses bear potential therapeutic prospect in creating a “good first impression” setting [28,29] for motivational and committed therapy or care process for adolescents. This will be done by connecting the elements of attachment bond to Doris Pierce’s model of therapeutically powerful activity, outlining three main sources of therapeutic power: (1) appeal, (2) accuracy, and (3) intactness [25]. 

### 1.1. Horses as Potential Attachment Figures

As large animals, horses are simultaneously frightening and attractive to many people. However, Wilson [12] has highlighted that horses are herd and prey animals, and thus, they are extremely aware of their environment and the intentions of other creatures in it. Horses have a well-developed “fight and flight” instinct, and for that reason, they are very sensitive to the body language and emotions that are unconsciously projected by humans [30,31]. When horses are healthy and not stressed, they approach humans with “cautious curiosity” [32], which differs from the familiar situation of an unfamiliar adult coming “too close too soon” and causing a child to feel intimidated and frightened [32]. Therefore, horses may be better than humans in respecting the space people need when getting familiar with other human and non-human beings [32]. Over recent years, the discussion surrounding the responses of horses to human emotional signals has increased. Horses have shown to be emotional beings, and there is clear evidence on horses’ ability to recognize human emotions and respond to them [33,34,35,36]. A horse will move away from a nervous person, because the situation causes him to feel anxious, but he will often come toward a calm person [37]. Horses tend to react to a person’s body language [31,38] and, according to Smith, Proops, Grounds, Wathan, and McComb [39], horses possess the ability to recognize the difference between positive and negative human facial expressions. Furthermore, Baba, Kawai, and Masahito [34] have shown that horses are sensitive to human emotional cues and modify their behavior based on the implied meaning of any negative or positive human emotional message. These outcomes have been supported by Proops, Grounds, Smith, and McComb [35], Smith [36], and Wathan, Proops, Grounds, and McComb [33], concluding that horses are capable of mirroring people’s emotions and responding to them appropriately in a given situation. This may be helpful for people who, when facing the horse, may not be able to recognize their emotional state. Thus, through acting as a mirror, a horse helps people identify their emotions and behavior [38,40,41,42].

Comparing dog–human and horse–human attachments, Payne, Dearaugo, Bennett, and Mcgreevy found evidence of the four principal criteria of the attachment bond—proximity maintenance, safe haven, secure base, and separation distress—in dog–human dyads, but it is still unclear and individual within horses if they may have people as attachment figures [43]. However, Lentini and Knox [30] suggest that horses may offer a more sensitive attachment figure than, for instance, cats or dogs because of the discrepancies between the animals’ nature. Dogs, although undeniably sensitive to non-verbal communication, by nature are predatory animals and have been throughout history accompanied humans as hunters, whereas horses are prey animals and have historically been the hunted ones [41]. Therefore, as a means of survival in the wild, horses have developed an exceptionally keen awareness of non-verbal communication, making them capable of offering immediate and unbiased non-verbal feedback to humans [40,42,43].

### 1.2. Horses in the Creation of Therapeutic Space: The Application of Pierce’s Model

Dunlop and Tsantefski [24] (p. 22) suggest that, “since the work of John Bowlby (1988), the availability of secure attachment relationships has been understood as crucial to infant and child development, as well as to social and emotional wellbeing into adulthood”. However, in their article on equine-assisted group therapy, they suggest that animals in therapy may be able to offer the experience of such an attachment, offering a relationship in which children feel safe, can experiment with extending themselves, and return for comfort and reassurance when their capacity is overwhelmed. This is supported by Vincent and Fargas [27], who argue that relationships between horses and humans can be studied in the context of attachment theory, and by Bachi [8], who suggests that Bowlby’s attachment theory can be used to explain the theory and practice of equine-assisted interventions. In this article, we investigate the potential development of attachment relationships between young people and horses according to the four principal features of attachment by Bowlby [13,15]. Additionally, we apply Pierce’s model of therapeutically powerful activity to study if the potential development of spaces and opportunities for attachment create more positive encounters between young people and care professionals. 

In her theory, Doris Pierce [25] outlines three main sources of therapeutic power. She argues that for people to be motivated, committed, and fully engaged in a therapeutically favorable activity, the situation needs to be “appealing”, “accurate”, and “intact”. 

Firstly, as one of the requirements for therapeutically motivating action, Pierce highlights the situation’s appeal. She argues that for clients to engage in the therapy process, the activities in the session must be attractive to them. According to Pierce [25], attractive activity feels desirable to clients and allows them to experience productivity, pleasure, and restoration. There is reason to believe that horses can provide an attractive and appealing environment for various therapy and care interventions through enabling multi-sensory experiences such as touch, emotion, physical and cognitive challenges, and the potential for learning new tasks [8,44,45]. 

Secondly, Pierce highlights the therapist’s or caregiver’s skills, naming them the “accuracy” of an activity [25]. Applied in the setting of equine-assisted interventions, this means that the caregiver or therapist needs to possess the appropriate skills to carry out a session, which involves horses. Russel-Martin [46] (p. 23) argues that a therapist has to know her/his rightful place between the client and the horse and, in particular, needs to be aware of what is happening between the client and the horse. The farm or horse stall settings bring with them specific challenges, such as other people and animals, restrictions on actions, animal behavior, and so on, which can hinder the therapist’s ability to maintain the client’s attention on the interaction with the horse and the therapist [47]. 

The third source of therapeutic power according to Pierce relates to the uniqueness of the context. Pierce calls this “intactness” and stresses the need for sensitivity for the particularities of the therapy context and the uniqueness of the session for the client [48]. Moreover, she highlights the naturalization of the therapeutic environment [25]. Horses, when involved in therapeutic or care interventions, provide particular spatial, temporal and socio-cultural conditions for a therapeutic encounter. Equine-assisted therapy carried out in natural surroundings enables a client to express different aspects of themselves. This is enhanced within the freedom of an open space, which can be relaxing and promote a more unrestricted expression of oneself [49]. 

Applying Pierce’s theory in the context of equine-assisted interventions highlights that it is not enough to consider only the attachment relationship between a person and an animal but also the whole environment needs to be taken into account. When animals are present, the environment is always particular for all those involved. The very essence of a horse offers both opportunities and challenges. It is expected that a relationship with an animal is an attractive proposition and can be a therapeutic experience [24]. This may be particularly significant in the early stages of trying to motivate adolescents to the therapy process, as animals can help to create a “good first impression” through promoting positive and hopeful attitudes, emphasizing the adolescent’s competence and advancing confidence in the therapy process [50]. Research findings have indicated that first impressions, even if quickly formed, are often accurate [28], and that clients’ first perceptions of therapists and therapy environments predict therapeutic outcome [29]. We assume that it is probable for the presence of horses to be able to assist in creating a positive, motivational, and committing atmosphere, which may enable adolescents to engage in processes that help them to thrive.

## 2. Materials and Methods 

In this article, we examine whether short-term exposure to horses, in this case only the presence of a horse, allow us to view horses as potential attachment figures and as animals holding potential in creating a therapeutically favorable environment.

### 2.1. Participants and Setting

The Youths. Nine adolescents, three boys and six girls 16–17 years of age, participated in the study. They were attending a tutorial course organized by the local church for becoming tutors for younger children in the church’s camps in the coming summer. They had participated in the church-related youth activities for one to two years and were acquainted with each other. They were used to having conversations with each other; hence, we expected a friendly and open atmosphere for sharing thoughts and feelings. As attendees of the voluntary training to become tutors, the participants may be considered as active young people who, as far as we are aware, have no history of special needs. Two of the girls had some experience with horses, and the others had not personally met with a horse before.

The Horses. Six horses were residents at the farm at the time of the study. For the session from which the study draws, the horses were divided into two groups. The first group consisted of a 23-year-old Norwegian Fjord horse gelding and two Irish cob mares, one sixteen years old and the other one five years old. In the second group, there was a 12-year-old Irish cob mare, her two-year-old filly, and a two-year-old Irish cob gelding. The horses were so-called “hairy horses” named after their hairy legs. Adult horses (more than five years old) were placid and steady by their temperament, and the two-year-old foals were lively and curious. The horses were checked once to twice a year by a veterinarian who specializes in horses, proving them to be in excellent health emotionally and physically. They have frequent contact with people of all age groups.

The Farm. The farm where the study took place provides equine-facilitated therapy as a public health service by two equine-facilitated therapists (a physiotherapist and an occupational therapist). The horses and the farm are inspected and accepted by the official health care service system of the province and by The Social Insurance Institution of Finland. As part of the therapy scheme, the Social Insurance Institution of Finland and the local hospital districts cover the payment commitments on behalf of the clients who have severe special needs and whose rehabilitation through equine- facilitated therapy is well justified. The participants of this study were not involved in a therapy scheme; instead, they were voluntary participants of a church-related tutorial course. In addition to the six horses, there were two dogs at the farm.

### 2.2. Ethics

The participating adolescents met with the researcher (Author 1) in a session where they were orally told about the study and the involved activity. This took place one week before the actual intervention. All the young people participating the tutorial course gave their written informed consent for inclusion in the study. The study was carried out in compliance with the ethical principles of research for human participants and ethical review in the human sciences in Finland. The horses’ welfare was carefully observed during the study, and they showed no signs of distress or uneasiness. During the session, the horses were free in the paddock, so they had the “freedom of choice” to approach or withdraw from the humans. Only one young person at a time entered the paddock with a youth worker who was familiar and competent with the horses. In case of any signal of potential harm, the farm staff were also prepared to remove the animals from the session. 

### 2.3. The EASEL Session

The intervention was based on the EASEL (Equine-Assisted Social and Emotional Learning) method. The EASEL method differs from the authorized equine-facilitated therapy in Finland; for instance, it is typical in equine-facilitated therapy that horses are handled, groomed, and ridden, unlike in the EASEL method [51]. In Finland, equine-facilitated therapy is accepted as an optional form of public rehabilitation for people who have severe special needs and, thus, it is sponsored by the government. However, it is legitimate to utilize the EASEL method in therapy.

The basic principle of the EASEL method is to emphasize the will of the horse. Typically, horses are kept free in the arena or on the field or in the paddock. During the EASEL session, horses are not led with rope or ridden. The activities are carried out unmounted. The first step is for the clients to observe a herd of horses and proceed to get familiar with them. It is central in the method to respect the horses and let the horses engage with people voluntarily. Then, getting to know the horses first is expected to lead to free play with horses, which is a central goal of the EASEL process. This requires the horses to have the right temperament to be suitable for EASEL activities, such as being calm and sociable [51].

The EASEL session for this study was part of a one-year course for adolescents who wanted to act as tutors in church camps. While the course prepares the adolescents for their tutorial task by training for different kinds of activities, plays, and helps support younger children as well as knowing how to find support for themselves, the purpose of this particular session with the horses was to offer the adolescents a moment to pause for reflection.

The church youth worker who arranged the session had attended the training in the EASEL method organized by the University of Jyväskylä, Finland. The session was part of his training. He wanted to utilize the EASEL method in his work with this group of adolescents because at this point of their tutorial training, it was thought to be useful to provide the group with an opportunity to learn about themselves and other members of the group. The principles of the EASEL were used in the implementation of the session in the following way.

The participants were met on the farm by the youth worker who asked them what their thoughts were about coming to the farm and whether they had been around horses before. The youth worker also told the participants that the session would provide them with an opportunity to imagine themselves in the role of group leader, and maybe by meeting the horses, they could find new ways of expressing themselves and learn something new about themselves. After the introduction, the youth worker and the young people were taken to observe the horses.

The participants were divided into two groups and asked to observe the horses in two different paddocks. Three horses were in each paddock. Although clear rules on safety when acting around horses were adhered to, the instructions given to the participants regarding their observing the horses were broad and open. The youth worker asked them to contemplate what kind of feelings horses evoked in them. The open-ended questions worked as an attempt to avoid influencing the adolescents’ thoughts, offering them an opportunity to discover something about themselves. The young people were told to stay quiet and patient when being near the horses so that everyone could have their space to observe the horses, listen to their emotions, and reflect on the situation.

The participants observed the horses from behind the fence in complete silence, without talking to one another. After spending approximately 10 min getting used to being close to the horses and starting to get a feel for how they behave, the two groups swapped places with each other and observed the other horses for a few more minutes. After that, they were told that they could now go into the paddock one at a time and, if they wanted, get close to the horses. The youth worker knew the horses well and was with the horses and the adolescents the whole time, ensuring a safe environment for everyone. Moreover, the horses were cold blood and therefore had a placid and gentle disposition. In addition, due to the horses’ participation in therapeutic work, they were used to various distractions such as other animals, different people, wheelchairs, tractors, and children’s toys. This allowed the young people and the horses to get better acquainted with each other in a safe environment.

Most but not all of the participants wanted to go into the paddock to get closer to the horses. The individual time spent with the horses varied among the participants; this part of the activity took approximately 30 min all together. Apart from the individual visits in the paddock, all nine participants wanted to stand by the fence and touch the horses. In the other paddock, there were two foals, of which one, in particular, was very curious. The participants were eager to pet that gelding foal from behind the fence. 

When everyone had finished observing the horses and those, who wanted to, had met the horses in the paddock, the adolescents and the youth worker regrouped behind the fence and talked about their experiences. The questions presented for open discussion were: What was it like being around the horses? What kind of thoughts or feelings did it raise in your minds when you saw the horses as a group? How did you feel about yourself when you were with the horses? After this immediate discussion, the group walked back to the stables and the discussion about their experiences continued in a friendly environment with a barbeque meal.

### 2.4. Analysis

After the observation session with horses, a free-formed group discussion was conducted with the youth worker and the participants in a relaxed setting as described above. The discussion focused on the participants’ feelings about horses, their thoughts of their experience of observing the horses in the group, and the horses’ behavior with other horses. This discussion lasted 60 min and it was recorded by the researcher (Author 1) with the consent of the participants. In addition, the discussions among the adolescents and the youth worker by the fence outside the paddock were recorded. The recordings were transcribed, which produced a dataset of 2885 words. The participants in the study were anonymized by the process, and pseudonyms are used in the presentation of results. The data analysis focuses solely on the expressions of the participating adolescents; the horses’ behavior is not the focus of this study. 

The analysis of the data was conducted by utilizing the NVivo12 QDA (qualitative data analysis) computer software, and it implemented the method of in-depth reading and two-level qualitative content analysis [52,53]. At the first level of the analysis, the transcripts of the participants’ discussion were coded to identify significant phrases that directly reflected the participants’ observations. The data-driven coding, performed by Author 1, was based on the adolescents’ expressed feelings, emotions, ideas, and associations, and on the features of horses expressed by the adolescents such as calmness, patience, sociability, beauty, their size, and their behavior toward the humans and other animals. Then, at the second level of the analysis, Author 2 was involved in reorganizing the data through theory-driven thematization. The transcriptions were organized based on the principal criteria of an attachment bond specified in Bowlby’s [14,15] theory of attachment. This meant categorizing the phrases that were identified in the first-level coding under the themes of (1) proximity maintenance, (2) safe haven, (3) secure base, and (4) separation distress, which according to attachment theory can be understood as an indication of an existing attachment bond [9,10,14,54]. From the total of 58 phrases that the participants produced, 43 phrases were categorized according to these themes. The remaining phrases concerned other things, such as discussing the tools that could be used in the camp next summer. 

In addition to the thematization of the attachment features, the data were also thematized according to Pierce’s concepts of (1) appealing (attractive), (2) accurate (skillful), and (3) intact (unique) activity to study the potential of a horse-based interaction in constructing a good first impression and a motivational space for (therapeutic) encounters of care professionals and young people. The phrases categorized according to themes of therapeutically powerful activity by Pierce were the same ones that fit in the categories of an attachment relationship. In what follows, we combine the two approaches—Bowlby’s attachment criteria and Pierce’s model for therapeutically powerful activity—to illustrate how a human–horse interaction may potentially promote the emergence of a motivational environment at an early stage of therapy or care process. To maximize the extent to which young people’s voices are accurately represented, verbatim quotes from the young people’s discussion are presented to illustrate the results.

## 3. Results

### 3.1. Proximity of Horses; Appealing Activity

Proximity—the first criteria of potential attachment [13,15]—of the horses prompted several affectionate comments from the young people. For example, they described the horses to be calm, patient, beautiful, lovely, nice, and social. When observing the horses behind the fence, a girl, Liisa, noticed how “*they [the horses] are so patient, they only stand still—well, I am not so patient*”. Another girl, Julie, saw the animals as “*lovely calm horses*”, and a third girl, Pirjo, mentioned how “*they are so beautiful*”. A large majority of the comments by the participants were related to the horses’ ability to attract them. The calmness and patience of the horses were the most common expressions made by the participants concerning the characteristics of the horses. In addition, the beauty and sociability of the animals were attractive to the adolescents. A girl called Paula, who had earlier experiences with horses, explained:
Always when you go near a horse, you feel sort of, a little bit, that you would like to stroke it.

In addition, some of the girls, who at first felt intimidated by the size of a horse, felt later encouraged to go into the paddock. They described their experience as follows:
I feel so small because they [the horses] are so big.(a girl, Pirjo)
I felt the horse acted like a magnet. Although I didn’t want to go to face the horse, I felt the horse was very attractive, and it made me want to face it.(a girl, Anne)

Even for the adolescents, whose first reaction to the horses was skeptical, the horses felt attractive and inviting. This was further illustrated by Anne’s narration of her encounter with the two foals. She felt particularly attracted by Snuffkin, the two-year-old gelding, who was a curious and sociable horse and who was always eager to get close to people. Anne explained meeting Snuffkin with the youth worker at the gate of the paddock:
So, first I thought they might eat me without a bite because they are so big and they are also so young still, and I saw earlier that they nibbled on each other. But instead, they were just nice, although another one (Snuffkin) was going to eat my handbag, but he was just as nice as the others because he came to say hello to me.

Although the horses were large and sometimes even intimidating to young people, they often present themselves so appealingly that people cannot resist trying to get to know them. Anne’s experience, as described above, confirms Bachi’s [55] argument that horses who have positive experiences with humans will usually seek the affection of a person, who caresses them, by getting physically closer. Attachment, in the form of proximity, was hence enhanced by the behavior of the horses. This was also noticed by Julie:
“Very sociable horses—came immediately over, came to see us, and I was like [expressing surprise] ‘uu uuu’”.

As highlighted by Pierce [25,48], a therapeutically motivating action needs to be appealing to the people involved. The adolescents participating in this study felt strongly drawn by the horses when allowed to be in proximity with them. This made the situation feel attractive and appealing. As horses provoke positive images in people’s minds, their presence will also reinforce positive interaction among the human beings present [55]. Based on the experiences of the adolescents in this study, it can be assumed that horses present a potential for making a therapy/care session attractive to a client. This promotes the meaningfulness of the session and the engagement of the client in the therapy process [25].

### 3.2. Safe Haven; Unique Situation

As described above, the adolescents commonly experienced the horses to be very tranquil. The feeling of tranquillity is an emotion that can be linked to the concepts of “safe haven” and “secure base” [9,19] which Bowlby names as illustrative elements of an existing attachment bond. Several participants expressed how the horses had a calming influence on them. For instance, Matti, Pekka, and Venla explained how the calmness of the horses made them feel serene:
I just, I was sort of like, I was calm when I watched the horses.(a boy, Matti)
They are so calm, I could be here for a long time just watching them. It’s sort of like they are standing and not doing anything… nevertheless, they look like they are doing something all the time.(a boy, Pekka)
At first I felt a little bit nervous, but after I was near the horses and one of the horses was eating, it was doing nothing, so it made me calm, it made the situation peaceful.(a girl, Venla)

According to Zilcha-Mano, Mikulincer, and Shaver [9], therapy pets may potentially become figures in a person’s attachment hierarchy. Based on the reactions of the participants in this study, it seemed that partly through the horses’ calmness, which provided a “safe haven” for the young people, they could act as sources of attachment for adolescents. Similarly, Dunlop and Tsantefski [24] in their study on an Equine-Assisted Therapy (EAT) program for children found that EAT offers an environment in which children can feel safe and secure. 

Horses can also present a “safe haven” for recognizing and expressing feelings and emotions. This was exemplified by a boy, Tommi, who originally said that he was afraid of horses. He felt that the horses were big and frightening and that he did not want to go to the paddock. “*Horses will bite and eat me,*” he said, and continued: “*they are so big, too.*” However, drawn by attractiveness of the horses and encouraged by the safe environment, he faced his fears and went into the paddock. He could not resist the “magnetism” of the horses. Afterwards, in the group discussion, he contemplated:
Yet I dared to go into the paddock. And one of them, a black horse [the 5-year-old mare] was so nice. She accepted me, she totally accepted me.

For Tommi, facing the horse seemed to be a healing experience. It seemed that his self-esteem was not great, but in the end, he was able to interact with the horse who gave him a feeling of being accepted. As a result, he felt that he is just like any other boy and more accepted for who he is. This example shows how horses can help humans in many ways deal with their feelings. In addition, as many of the participants felt compelled to stroke the horses, it seems that the calm presence of horses may potentially increase empathy in young people. Through their gentleness and tranquillity, horses may provide a unique environment. This is identified by Pierce [25] as one of the requirements for a therapeutically powerful activity. Pierce highlights the contextual dimensions as important sources of a successful therapy session. Thus, it can be assumed that horses have the potential to be meaningful elements in creating “intactness” as part of a therapeutically favorable atmosphere. 

### 3.3. Secure Base; Accurate Environment

Tommi’s experience, as narrated above, can also be seen as an example of how horses may provide a secure base—the third of Bowlby’s four criteria of an established attachment—for adolescents. As in the presence of the horse Tommi felt accepted, he gained more confidence in interacting with the people around him as well. Julie, who had been around horses earlier, agreed:
The good side of a horse is that it does not think bad of you, or evil things about you.

As adolescents are often preoccupied with how others see them, it becomes increasingly important to recognize the unchallenged approval of an animal, in the presence of which they do not need to be ashamed of the way they behave, feel, or look. For instance, Bachi [55] has found that horses, in particular, are non-judgmental and that they project love and acceptance to whoever treats them positively. Julie explains how this provides also a lesson on how to approach other humans and non-humans:
It was also good that the horses slowly get used to you as they approach you. And if you are not in a good mindset, then the animals will easily run away if you’re not in a good mindset, they won’t obey. But if you go near them when you are calm, you can work better with them.

It was observable from the participants’ comments that horses may help create a secure base for adolescents from which exploring the surrounding environment and relationships feel safe. However, this requires skillful guidance from the instructor. According to Pierce, this in a therapy/care setting may be conceptualized as “accuracy”, referring to a therapist’s skills to design and carry out a successful session [25]. Particularly when animals are involved, the role of a worker or therapist is of crucial importance. The client needs to possess a high level of trust toward the care professional to feel safe around a horse. This requires a good relationship between the professional and the horse. In the group discussion, one of the boys, Pekka, expressed his thoughts about being in the paddock with the horses and the youth worker:
I didn’t get nervous, because I saw that you [speaking to the youth worker] were so confident with the horses. I couldn’t help facing the horses. I just trusted you.

In addition to the client becoming more confident and trustful in the presence of horses and a skillful care professional, horses may also increase the professional’s ability to intervene. Zilcha-Mano [9] shows how a therapist may draw motivation and friendship from an animal to face the client in an open atmosphere. Horses may also help the therapist feel relaxed, which allows him to be more emotionally positive than without the presence of horses. Hence, a horse’s role in a therapy/care session may facilitate the therapists’ skills and the “accuracy” of the activity [25].

### 3.4. Separation Distress; Space for Self-Reflection

According to attachment theory, separation distress is experienced when the attachment figure is temporarily unavailable [15]. Some features of separation distress, as described by Bowlby [13] and others [15,18,56], were identified also in the phrases of our participants. Especially the 16-year-old mare, who according to the participants seemed unwilling to get familiar with people, evoked feelings typical to separation distress. Venla relates:
She could just have... so that she was in that mood where she didn’t feel anything, and if someone would have come too near to her, she may have sensed something which didn’t feel so good. She may have thought that there was a threat, or that it was in her nature to be a little bit... If someone new would come to see her for the first time, she might wonder if they are coming to poke her. Well, she is a little bit independent.

While the participants could empathize with the less sociable horse, many of them felt disappointed with the horse’s withdrawal. An animal that they felt very drawn to showed only a little interest in them. Julie compared the mare’s behavior to that of a human being:
It can also be the sort of feeling that if someone, a type of person who is not at all interested in you or anyone else, and if someone gets too close to that person, it would be like he would be in his own bubble where nobody is not allowed to step in. It is just those horses, for example that horse, who was eating alone. So it raised in me the feeling that if I would be eating in peace, I wouldn’t like to be disturbed either when I am eating.

Julie saw the connection between human and animal behavior in the situation where a person, whom one has an interest in does not want to be connected to them. In this kind of situation where one has somebody who they feel drawn to paying no attention to them, it is not atypical to feel disappointed. The feeling was also shared by Matti:
Matti:The other one [of the horses] looked at us for quite a long time, it looked for, sort of a long time.
Youth worker:Could you explain more?
Matti:So I wondered why I am still standing here.

In Matti’s image, the horses were “*just eating and slobbering*” and did not care for him or the others. As notified in the attachment theory [15], disappointment is the feeling that one gets when sensing the distance and receiving no attention from the object of their desire. As in Matti’s narration, Tommi also expressed anxiety caused by the indifferent horse:
… and then I felt a little bit that [the horses] first looked at us and they noticed that we are here and we are going to remain here for a little while, it was sort of like, to them that we would be like air to them.

The feeling of invisibility or not meaning anything to someone who she/he feels attracted to might be a strong feeling and cause (separation) anxiety, as stated in the attachment theory [13]. 

Related to the experiences of separation distress, Venla talked openly in the group discussion about the insights she had gained during her visit to the farm. She explained that she had learned something new about her behavior toward other people. The open and tactile atmosphere with the horses encouraged her to express things about her inner world:
I have one thing on my mind … It was something I didn’t understand at first, or something I have maybe only partly noticed before. When meeting people or animals, regardless of whether it’s people or animals, I feel I am a little reserved. It was then when the horse headed off toward the shelter in the paddock. So then at that moment, I didn’t want to go near the horse, I didn’t want to be … I didn’t want to find myself being made the center of attention.

When the horse turned her head away from Venla, she sensed what is defined as another sign of separation distress in the attachment theory: the futility of making contact to an attractive object in the expectation of one showing no interest and potentially abandoning you [15]. As the horse may have sensed Venla’s caution coming nearer, she withdrew, which caused distress in Venla. However, Venla was able to understand the horse’s caution, which was similar to her own. This kind of experience may be assumed to occur especially in the proximity of horses; if the animal had been a dog, it would have probably approached Venla directly despite her caution [26]. Venla came forward with her experience in the group discussion with the other young people:
When I was first brought to this place, I didn’t even want to greet or pet the dogs in the garden because I felt too shy. After these experiences, I suddenly realized that in new situations, I am always too nervous to meet new people or animals. But after being in that new situation for a while, I gradually dared to reach out to others. I think it has been the same way with you guys, and I realized it just now when I met the horses in the paddock. Maybe other people think that I don’t want to get familiar with them because I am a little bit reserved, but it takes time for me to face others. After being with others for a while, I start to open up to people and let them in.

Venla’s narration indicates how by humanizing horses, people may recognize and express their thoughts and views about themselves and their relationships with other people. Consequently, the presence of horses can make it easier for adolescents to reflect on themselves in the care process.

## 4. Discussion

This article investigated the potential of adolescents developing attachment relationships with horses as described in Bowlby’s attachment theory. Furthermore, we have applied Pierce’s model of therapeutically powerful activity to study if the emerging attachment encourages the development of early-stage encounters between care professionals and young people. In the analysis, we have looked at what kind of feelings, emotions, ideas, and associations being around horses evoked in the participants, and we studied their narrated experience in terms of the attachment theory and the favorable therapy/care environment for adolescents.

The results indicate that being in the presence of horses even for an individual short-term session may create a peaceful atmosphere, bring pleasure, encourage the expression of feelings, provide reinforcement and motivation, and encourage bonding with other humans and non-humans. Based on the discussions with the participants and their youth worker, the human–horse bond incorporates the four typical features of an attachment relationship. The clearest perceived attachment feature among the participants concerning the horses was the appealing presence of the horses, which was established in the proximity of the animals. Horses also had a calming effect on the young people, pointing to the attachment criteria of offering a “safe haven”. In addition, with the help of a skillful care professional, the environment created through the horses’ presence generated a “secure base” for the adolescents. Furthermore, the study indicates that a human–horse bond may also bear signs of the fourth criteria of attachment, namely separation distress. Some of the participants’ comments can be identified as indicators of separation distress, although separation distress as a feature of an attachment bond is expected to occur mainly when the attachment is deeper and established in lasting psychological connections [13,14].

The results substantiate the findings by Dunlop and Tsantefski [24] who, in their study with younger children with special needs, found horses to be creatures who provide feelings of safety and security and enhance personal and social development. Accordingly, based on her study on young self-harming female adolescents, Carlsson [57] suggests that horses have a calming effect and that they provide adolescents with non-verbal and non-judgmental feedback on their emotions. The equine-assisted intervention implemented in Carlsson’s study boosted the clients’ feelings of trust, patience, and empathy, and they provided a “moment of silence” for the clients [57]. Similar phenomena were also found in the study by Hemingway, Carter, Callaway, Kavanagh, and Ellis [44] in their study of the mechanism of action of an equine-assisted intervention.

Furthermore, this article looked at the potential therapeutic elements in the one-off session with the adolescents who were participating in a tutorial course organized by the local church. While Pierce’s model, which our investigation draws from, is created for therapeutic practice, the participants of this study were “ordinary” young people with no history of special needs. Despite this, we claim it fruitful to study the participants’ experiences from the viewpoint of promoting attachment and establishing a motivating environment for early-stage professional care encounters. In the purpose of finding out fundamental information about human–animal bonds, it is worth investigating also the perspectives of adolescents without special needs. The basic information gained in “snapshot” studies such as ours can help us understand the rules of attachment when aiming for therapeutic alliance in human–animal-assisted interventions and when dealing with people who have difficulties in forming an attachment.

Nevertheless, despite the snapshot nature and the limited number of the participants in the present study, the results support the argument that there are motivational elements even in a short one-off session involving horses. These elements may help to create a good first impression, which was found to be of vital importance in the beginning stage of a therapy or care process with adolescents [29]. Based on their experiences of being around the horses for a short time and, for most of them, for the first time in their life, the participants expressed thoughts that can be interpreted through Pierce’s concepts of therapeutically powerful activity. Firstly, the horses appeared as attractive and appealing to the adolescents. Secondly, the context was unique and intact for the participants. Finally, in the group discussion, the participants conveyed the trusted role of the youth worker when around the horses, indicating—by using Pierce’s terms—an accurate environment. Additionally, the presence and interaction with the horses provided space for self-reflection. Therefore, as the participants’ narrated experiences involved elements of “therapeutically powerful activity”, we suggest that horses are animals that can create a positive first impression and a therapeutically favorable space for adolescents. 

It is generally assumed that a good therapeutic relationship between a client and a care professional is the most powerful factor in a therapeutic process [58]. Moreover, the therapeutic alliance is perceived to be closely linked to the attachment between the client and the professional [58]. As research has shown, the emotional bond between humans and animals has similar features to the emotional bond among humans [8,9,26]; thus, a therapy animal can “attract a client like a magnet”, creating a strong attachment between the client and the animal [59]. We conclude that based on the results of this study, horses may be seen as sources of emerging attachment to adolescents and that they have the potential to enhance young people’s motivation for engaging in a favorable care situation, which is shared with horses and other people.

To conclude, we wish to stress that while the research in the field of human–horse attachment is increasing, it is still reasonably uncommon. More research efforts are needed in areas such as human–horse attachment but also in horse–human dyads. It is important to know what humans mean to horses and whether a human can be an attachment figure to a horse. For instance, we know very little about what attracts and activates the interest of horses toward humans and how individual these features may be in horses and in people [18,60]. As Payne, Dearaugo, Bennett, and Mcgreevy [18] highlight, at this stage, we are only at the start of the discussion on the topic. Furthermore, our study does not provide information on, for instance, how the initial feelings of the participants may change in time if they associate more with horses along the therapy or care process or if they learn more about horses as prey animals. Neither does it consider the other types of bonding that adolescents may develop in the process that may also be beneficial for them. In the future, learning about both sides of the human–horse dyads in more long-term research interventions would help us interpret the mutual interaction better. This will enhance the versatile involvement of horses as therapy and care companions.

## 5. Conclusions

Given the rise in equine-assisted interventions in recent years, it is important to inquire what are the mechanisms that make horses create an effective environment for therapy or professional care. It has been suggested that for children and young people, an attachment bond between them and an animal may work as such a mechanism. Earlier research has presented evidence of attachment bonds in dog–human dyads, but it has to date been unclear if people may have horses as attachment figures. This study investigated the potential emergence of an attachment relationship in human–horse interaction. Further on, it was studied whether horses may be regarded as animals with potential therapeutic prospects through their ability to assist in creating a motivational environment for early stages in a therapy or care process. 

As its theoretical basis, the study drew from Bowlby’s attachment theory and Pierce’s model of therapeutically powerful activity. Based on a “snapshot study” of a time-limited session with three boys and six girls, aged 16–17, who observed and encountered free-roaming horses at a farm, and an open-ended group discussions following that, the two-level qualitative content analysis looked at the feelings and first impressions that the horses evoked in the adolescents when they observed and interacted with the horses for the first time. The results suggest that horses may activate the emergence of attachment in young people in the same way people do, which was illustrated by elements of the four principal criteria of an attachment bond: proximity maintenance, safe haven, secure base, and separation distress. These elements of attachment intersect with the features of a therapeutically favorable activity, namely the session being perceived as appealing, unique, and accurate by the young people. Hence, the study offers preliminary information about horses’ therapeutic and care potential for dealing with adolescents with and without special needs. The study indicates that in equine-assisted care work with adolescents, elements of attachment that occur in an appealing, unique, and accurate setting may promote successful interventions, hence suggesting that the attachment theory combined to the model of therapeutically powerfully activity may give a good structure to the equine-assisted therapy sessions and increase the theoretical understanding of the presence of horses in therapy and care sessions. However, future studies are needed to explore more deeply the dyads between adolescents and horses, also including the horses’ experience, and the further development of a potential attachment between horses and humans and its significance in reaching the expected outcome in the therapy and care of adolescents. This will require the possibility for research participants to develop a relationship with horses for a longer period of time.

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
