# Peer review of "The Potential of Human–Horse Attachment in Creating Favorable Settings for Professional Care: A Study of Adolescents’ Visit to a Farm"

_animals, 2020, doi:10.3390/ani10091707_

Round 1

Reviewer 1 Report

The idea of the study is interesting but the short -time spent with the horse, and the relationship only based on its observation and a very small contact doesn't appear sufficient to stress that the attachment sistem is activated. Moreover there is not a control group in the study and the results are discussed on the basis of a qualitative analysis.

The study could include more interactions with horses, in different situations (for example grooming) for a longer time

Author Response

Dear Reviewer 1,

We are pleased to resubmit the manuscript after making the required revisions. We thank you for your helpful comments that clearly improved our manuscript. We have addressed the comments as listed below. The changes can also be seen in the manuscript through the ‘Track Changes’.

Reviewer: The idea of the study is interesting but the short -time spent with the horse, and the relationship only based on its observation and a very small contact doesn't appear sufficient to stress that the attachment sistem is activated.

We have clarified the design of the study and the research question by stating more clearly than in the earlier version that in our study we do not wish to claim that the attachment system is activated in short-time session but that we are studying the potentially emerging attachment to horses which adolescents may develop when they meet horses for the first time. We have added relevant references to support this thought (see refrences: [27-28; 48]). (See e.g. lines 12-15; 25-26; 33, 82-87; 175-180, 533-535, 575-576, 600-605, 623-625)

Reviewer: Moreover there is not a control group in the study and the results are discussed on the basis of a qualitative analysis. The study could include more interactions with horses, in different situations (for example grooming) for a longer time.

The methods section of the paper has been revised throughout (see chapters 2.3 and 2.4). The study design and the analysis are illustrated in more detail in the revised manuscript. In our field (social work, youth research and social sciences) this kind of a small-scale research design is not atypical.

In addition, the revised manuscript has gone through an additional language check by a native English speaker.

Dear Reviewer, Please find enclosed our manuscript version with track changes

Reviewer 2 Report

Thank you for your interesting paper. I just have a few questions and suggestions.

I’m not sure about the title, would it be better as “human – horse attachment” as you state you have made no measures of the horse and you use human-horse in your text mostly.

Line 89 – would “big” be better replaced by “large”?

Line 91 – 92 – I think you need a citation that supports both the eagerness of horses to approach humans and another to support the intimidation felt by some humans as this help lend credence to your points

Line 103 – you use the term “trusts” here in the term that the horse will follow someone they trust, this is a pretty contentious issue in horse training and not really supported by all the science. Would it be better to avoid the “follow” and “trust” term and just state that the horse’s behaviour will be calmer around a calm human? There are some nice papers out there, one by Keeling et al. (2009) which investigates the impact of a nervous human on horse behaviour which may be of interest, in The Veterinary Journal. That may give some better terms to use than “trust”.

Line 120 – I think this could maybe be reworded a little as it sounds a bit like you are suggesting dogs are not sensitive to non-verbal communication.

Line 173 – How was informed consent gained from under 18 participants? You mention that this was given later, but did you send them a written information pack or give them a briefing?

Line 205 – You mention earlier that all the horses were in excellent health, emotionally and physically and then that they are inspected once or twice a year by an equine vet. Is this really enough to say that their well-being is excellent at the time of your study? Were there any exclusion criteria for the horses e.g. did you have a list of behaviours from them that would cause you to then take them away from the session such as distress, lameness etc.?

Line 219 – Are horses really able to “make friends” with us?

Line 220 – Are there any safety issues in this free play? How are the children’s safety ensured with loose horses?

Line 225 – Is one supervisor enough in a session with this many participants and this many free horses in two paddocks?

Line 282 – Were your data only analysed by one researcher or were the data triangulated at all?

Line 307 – Quote not in italics here as per the rest of your quotes

Line 318 – not sure I understand what you mean by “were not merely positive”?

In the discussion you make some good points about the need to also study the horses’ experience, I think you could make a little more of this for suggestions for further research.

Author Response

Dear Reviewer 2.,

We are pleased to resubmit the manuscript after making the required revisions. We thank you for your helpful comments that clearly improved our manuscript. We have addressed the comments as listed below. The reviewer’s original comments are in black and our responses follow in red. The changes can also be seen in the manuscript through the ‘Track Changes’.

Reviewer: I’m not sure about the title, would it be better as “human – horse attachment” as you state you have made no measures of the horse and you use human-horse in your text mostly.

We have corrected this very clear mistake.

Line 89 – would “big” be better replaced by “large”?

We have replaced “big by “large”.

Reviewer: Line 91 – 92 – I think you need a citation that supports both the eagerness of horses to approach humans and another to support the intimidation felt by some humans as this help lend credence to your points

A mistake in these sentences, that happened in the translation process from Finnish to English and caused a misunderstanding, has now been corrected. We have added a reference to support the original idea. (See lines 97-100.)

Reviewer: Line 103 – you use the term “trusts” here in the term that the horse will follow someone they trust, this is a pretty contentious issue in horse training and not really supported by all the science. Would it be better to avoid the “follow” and “trust” term and just state that the horse’s behaviour will be calmer around a calm human? There are some nice papers out there, one by Keeling et al. (2009) which investigates the impact of a nervous human on horse behaviour which may be of interest, in The Veterinary Journal. That may give some better terms to use than “trust”.

We have corrected the sentence in the suggested manner. (See lines 104-105.)

Reviewer: Line 120 – I think this could maybe be reworded a little as it sounds a bit like you are suggesting dogs are not sensitive to non-verbal communication.

We have revised the wording on this sentence to acknowledge the sensitivity of dogs to non-verbal communication. (See lines 121-124.)

Reviewer: Line 173 – How was informed consent gained from under 18 participants? You mention that this was given later, but did you send them a written information pack or give them a briefing?

We have explained the consent procedure more carefully. (See lines 225-226.)

Reviewer: Line 205 – You mention earlier that all the horses were in excellent health, emotionally and physically and then that they are inspected once or twice a year by an equine vet. Is this really enough to say that their well-being is excellent at the time of your study? Were there any exclusion criteria for the horses e.g. did you have a list of behaviours from them that would cause you to then take them away from the session such as distress, lameness etc.?

The text regarding the well-being of the horses has now been clarified. (See lines 206-208; 230-234.)

Reviewer: Line 219 – Are horses really able to “make friends” with us?

We have replaced "make friends" with "get to know". (See line 244.)

Reviewer: Line 220 – Are there any safety issues in this free play? How are the children’s safety ensured with loose horses? (AND) Line 225 – Is one supervisor enough in a session with this many participants and this many free horses in two paddocks?

We have explained more carefully how the young people’s safety was ensured during the session. (See lines 228-230)

Line 282 – Were your data only analysed by one researcher or were the data triangulated at all?

We have revised the text regarding the analysis of the data, making it clearer that both authors participated in the analysis of the data. (See Lines 310-318)

Line 307 – Quote not in italics here as per the rest of your quotes

We have corrected this.

Line 318 – not sure I understand what you mean by “were not merely positive”?

The expression "not merely positive" was replaced by "sceptical".

Reviewer: In the discussion you make some good points about the need to also study the horses’ experience, I think you could make a little more of this for suggestions for further research.

We have included the idea of studying horses’ experience as a suggestion for further research. (See lines 636-639.)

Furthermore, the methods section of the article has been revised throughout. In addition, the revised manuscript has gone through an additional language check by a native English speaker.

Dear Reviewer, Please find enclosed our manuscript version with track changes.

Reviewer 3 Report

This paper explores the possible ways to conduct horse-human therapies, and is a pilot type of study which could lead to more elaborate studies.

Authors uses Bowlby's four factors to analyze the comments by the participants.  Bowlby's attachment theory is based on mother-infant relationship which is a lasting psychological connections.  Caregivers provide safety and security to the infant.  

I am not sure if the horse human bonds can be compared to Bowlby's attachment.  As the authors mention, horses are prey animal that they are not most fit type of animal to provide safety and security.  The term attachment can be used in human and horse relationship in terms of human feeling close to the horse, but NOT in Bowlby's sense of attachment.

The initial feelings the participants felt may change as they get to know the horses or they learn about the prey animal.  They may develop other type of bonding, which is also good for them.

I recommend that authors use other definitions or terms to explain the emotional closeness the participating youth felt after observing the horses.  

One methodological question.  How long did youth spend observing the horses after the initial 10 minutes.(line 262)

Author Response

Dear Reviewer,

We are pleased to resubmit the manuscript after making the required revisions. We thank you for your helpful comments that clearly improved our manuscript. We have addressed the comments as listed below. The changes can also be seen in the manuscript through the ‘Track Changes’.

Reviewer:

This paper explores the possible ways to conduct horse-human therapies, and is a pilot type of study which could lead to more elaborate studies.

Authors uses Bowlby's four factors to analyze the comments by the participants.  Bowlby's attachment theory is based on mother-infant relationship which is a lasting psychological connections.  Caregivers provide safety and security to the infant.  

I am not sure if the horse human bonds can be compared to Bowlby's attachment.  As the authors mention, horses are prey animal that they are not most fit type of animal to provide safety and security.  The term attachment can be used in human and horse relationship in terms of human feeling close to the horse, but NOT in Bowlby's sense of attachment.

The initial feelings the participants felt may change as they get to know the horses or they learn about the prey animal.  They may develop other type of bonding, which is also good for them.

I recommend that authors use other definitions or terms to explain the emotional closeness the participating youth felt after observing the horses.  

In the article we present several examples of previous research that has used Bowlby’s attachment theory when studying the interaction between humans and animals, and we have added references justifying our research task, the design of the study and our choice of concepts (see references  [e.g. 2,8-9, 10, 11, 18, 24, 26, 27, 40,57]) and revised the text to state more clearly that we do not wish to claim that the attachment system is activated in the young people but that we are studying the potentially emerging attachment to horses which adolescents may develop when they meet horses for the first time. We have added relevant references to support this thought (see references: [27-28; 48]). (See e.g. lines 12-15; 25-26; 33, 82-87; 175-180, 533-535, 575-576, 600-605, 623-625)

Reviewer: One methodological question.  How long did youth spend observing the horses after the initial 10 minutes.(line 262)

We have explained the methods more precisely throughout (see chapters 2.3 and 2.4)  and answered this specific question on line 286 .

Overall, we have revised the introduction, clarified the research design, explicated the methods more carefully, and revised the discussion to present the results and conclusions more clearly. In addition, the revised manuscript has gone through an additional language check by a native English speaker.

Dear Reviewer, Please find enclosed our manuscript version with track change.

Round 2

Reviewer 1 Report

Thank you per improving the information. The paper is appropriate.

Reviewer 3 Report

With the revisions authors have made, the importance of horses as mediator to therapeutic situation.  The results, even though a short session, could help others to use horses as an introductory tools to start up professional cares with clients who may be hesitant in getting involved.

I hope authors extend their study to provide more evidence to convince psychologists to use horses as part of their programs.